# Atomistic Origins of Resurrection of Aged Acetylcholinesterase by Quinone Methide Precursors

**DOI:** 10.3390/molecules29153684

**Published:** 2024-08-03

**Authors:** Leonardo V. F. Ferreira, Taináh M. R. Santos, Camila A. Tavares, Hassan Rasouli, Teodorico C. Ramalho

**Affiliations:** 1Laboratory of Computational Chemistry, Department of Chemistry, Federal University of Lavras, P.O. Box 3037, Lavras 37200-000, MG, Brazil; leonardo_lavras@hotmail.com (L.V.F.F.); tainah-martins@hotmail.com (T.M.R.S.);; 2Center for Basic and Applied Research, Faculty of Informatics and Management, University of Hradec Kralove, 500 03 Hradec Kralove, Czech Republic; 3Medical Biology Research Center (MBRC), Kermanshah University of Medical Sciences, Kermanshah 6714414971, Iran; h3n.rasouli@gmail.com (H.R.)

**Keywords:** nerve agents, acetylcholinesterase, resurrection, realkylation, mechanistic studies

## Abstract

Nerve agents are organophosphates (OPs) that act as potent inhibitors of acetylcholinesterase (AChE), the enzyme responsible for the hydrolysis of acetylcholine. After inhibition, a dealkylation reaction of the phosphorylated serine, known as the aging of AChE, can occur. When aged, reactivators of OP-inhibited AChE are no longer effective. Therefore, the realkylation of aged AChE may offer a pathway to reverse AChE aging. In this study, molecular modeling was conducted to propose new ligands as realkylators of aged AChE. We applied a methodology involving docking and quantum mechanics/molecular mechanics (QM/MM) calculations to evaluate the resurrection kinetic constants and ligand interactions with OP-aged AChE, comparing them to data found in the literature. The results obtained confirm that this method is suitable for predicting kinetic and thermodynamic parameters of ligands, which can be useful in the design and selection of new and more effective ligands for AChE realkylation.

## 1. Introduction

Chemical warfare agents (CWAs) are toxic chemicals employed in military operations to inflict casualties (e.g., death, injury, and incapacitation) [1,2]. These agents, encompassing gaseous, liquid, and solid forms, pose a significant threat due to their potent toxicity towards humans, animals, and plants [1,2].

Following the horrors witnessed in World War I, the international community recognized the devastating consequences of CWAs. To prevent their future use, the Chemical Weapons Convention (CWC) was established in 1993 [3]. Administered by the Organization for the Prohibition of Chemical Weapons (OPCW), the CWC has played a crucial role in eliminating existing stockpiles of CWAs [3,4,5].

Despite a vast array of known toxic substances, only a select few meet the stringent criteria for CWA classification, including high toxicity, rapid action, and persistence in the environment [3]. Nerve agents (NAs), a particularly lethal class of CWAs, were developed during World War II [6]. Notably, NAs share a structural similarity with organophosphorus compounds (OPs), which are widely used in various applications such as in agriculture (pesticides), industry (softening agents), and medicine (cholinesterase inhibitors) [6]. This widespread use of OPs, particularly in agricultural settings of developing nations, contributes to the concerning global burden of OP poisoning [6]. An estimated 3 million cases of OP poisoning occur annually, with accidental contamination due to misuse and intentional self-harm as leading causes [7,8]. These cases translate to a staggering 260,000 deaths per year [7,8]. Beyond their use in warfare, NAs have been employed in terrorist attacks, with documented incidents involving sarin (GB) and Tabun (GA) during the 1980s Iran–Iraq War [9]. The threat of NAs extends to more recent events, including the 1994 and 1995 sarin attacks in Japan and the 2017 VX assassination of Kim Jong-nam in Malaysia [9].

The pervasiveness of OPs in agriculture, coupled with the lingering stockpiles of NAs, necessitates ongoing vigilance regarding the global threat of OP poisoning. The potential for accidental exposure during excavations and construction activities further underscores this concern [10,11,12,13]. Given the severity of OP poisoning, research efforts directed towards improved treatment strategies remain paramount.

Organophosphate (OP) compounds exert their toxic effect through the irreversible inhibition of cholinesterase (ChE), particularly the enzyme acetylcholinesterase (AChE) [14]. AChE is a critical enzyme responsible for hydrolyzing the neurotransmitter acetylcholine (ACh) at the neuromuscular junction and within the central nervous system, thereby regulating synaptic transmission [14]. The current treatment protocol for OP poisoning involves a two-pronged approach:

Muscarinic Antagonists: These drugs, such as atropine, competitively bind to muscarinic receptors, blocking the excessive stimulation caused by ACh accumulation and mitigating the associated symptoms [10,15,16].

Oxime Reactivators: Compounds like pralidoxime can reactivate inhibited AChE by cleaving the phosphyl group bound to the enzyme’s active site via a nucleophilic attack. This restores AChE function and reduces the toxic effects of OPs until they are degraded, and AChE levels return to normal [10,15,16].

Although reactivation of inhibited AChE has proven to be a challenging task over the past 70 years, the process is further complicated by the competition between aging and reactivation rates [17,18,19]. Aging is the spontaneous dealkylation of the alkyl group (R) from the phosphorylated serine residue (inhibited), resulting in an anionic phosphorylated serine. This aging process removes the alkyl side chain from the original OP, rendering AChE irreversibly inhibited (Figure 1) [20]. Despite numerous attempts and decades of effort, aging was considered irreversible, and until 2018, there were no reports of any in vivo or in vitro recovery of aged AChE activity.

Although some alkylation strategies were effective in model systems, these attempts, according to the literature, proved unsuccessful in vitro with aged AChE [21,22]. However, in 2018, Zhuang and colleagues published the first report demonstrating the in vitro resurrection of aged AChE activity using a small drug-like compound. This process is referred to as “resurrection” because the aged (inactive) form of AChE is restored to its active native form [23]. In this work, they focused on a class of compounds called “quinone methide precursors” (QMPs). These compounds, in aqueous media, generate an electrophile upon expulsion of a leaving group, which rapidly captures nucleophiles in the vicinity. The QMP called C8 (Figure 2) appeared to fit perfectly into the active site of two aged AChEs, generated by the reaction of Electrophorus electricus AChE with diethylfluorophosphate and with an analogue of soman. Both showed resurrection activity with interaction with C8 for 4 days (20.4% and 32.7%, respectively) [23].

Additionally, it was discovered that C8 (and other QMPs) are also reactivators of OP-inhibited AChE. Thus, QMPs offer the possibility of developing an antidote that not only alkylates aged AChE (the resurrection reaction) but also displaces the phosphoryl moiety from the active site nucleophilically (the reactivation reaction, similar to oximes). Although remarkable levels of resurrection were achieved in this work, the reaction is still too slow to be medically useful [21].

If the QMP reaction with aged AChE can be made faster and more effective, the development of a satisfactory antidote would be attainable. This research has paved the way for further studies, and the use of computational chemistry combined with in vitro analyses is a promising alternative in the discovery of new ligands capable of realkylating aged AChE [21]. This work employed a hybrid quantum mechanics/molecular mechanics (QM/MM) approach to investigate the resurrection of QMPs in enzyme–OP complexes. By utilizing molecular docking poses and QM/MM calculations, researchers were able to determine the resurrection energy barrier for each enzyme–OP complex with the QMPs. This not only validates the effectiveness of the QM/MM approach but also provides detailed insights into the reaction pathway and energy barriers involved. This paves the way for further development of QMPs as potential antidotes for OP poisoning. Unfortunately, to date, there is no effective “resuscitator,” meaning a broad-spectrum ligand capable of resurrecting OP-inhibited aged AChE [24].

Therefore, considering the importance of structural parameters for the efficiency of quinone resurrection, the aim of this study is to evaluate and compare the influence of these ligands (Figure 2) on the percentage of resurrection of AChE aged by the OP soman.

## 2. Methodology

### 2.1. Docking Study

For the docking studies, the crystallographic coordinates of acetylcholinesterase aged by soman were used, available in the RCSB Protein Data Bank (PDB code: 2WG1). To optimize the calculations, the aged AChE enzyme was prepared using Discovery Studio software [25], which allowed the removal of crystallographic water molecules distant from the active site, the calculations of the charges, and the addition of hydrogen atoms.

Regarding the ligands, their structures were previously constructed based on the information obtained from the work developed by Zhuang et al. (2018) [23], using the Avogadro program. The 2D structures of each ligand are presented in Figure 2. Subsequently, the molecules were optimized using Density Functional Theory (DFT), employing the B3LYP density functional and the 6-311 + G** basis set in CPCM (water) solvent. The partial charges of the atoms were calculated using the MK method, employing the Gaussian 09 package.

The present study performed docking calculations using the Molegro Virtual Docker Docker 2006® (MVD®) software [26]. This software utilizes scoring functions that estimate the binding energy of the ligand–receptor complex, taking into account various factors such as intermolecular interactions, solvation, and entropic effects. MVD employs the MolDock algorithm, which is based on scoring functions derived from the force field. These scoring functions result from the summation of contributions from bonded terms (bond stretching, angle bending, and dihedral variation) and non-bonded terms (electrostatic and van der Waals) into a principal overall function. MolDock utilizes a heuristic search algorithm that combines cavity prediction algorithms and differential evolution. Its scoring function is derived from the piecewise linear potential [26].

Molecular docking was used to predict the interaction modes between the aged AChE and the studied ligands (QMPs), aiming to discover the best poses of the ligands in the cavity of the enzyme. The parameters used to obtain the docking structures include atom volumes and cavity volumes, as well as the charges associated with the atoms involved. In this study, a total of thirteen molecular docking calculations were performed, with the first one used to validate and standardize the methodology, also known as re-docking. The ligand 534_FP1065 (LA), present in the crystal structure 2WG1, underwent the re-docking process with a spherical restraint of 10 Å. The scoring function used was Moldock score [GRID] with a resolution of 0.3 Å, and the algorithm employed was MolDock Optimizer with 100 runs, constrained within 9 Å, and residues within a 12 Å distance from the 79.86 cm^3^ cavity in the enzyme’s active site. The ligand–enzyme system was considered flexible for all calculations. In flexible docking, all species of the complex are kept free, allowing them to adjust their positions to optimize interactions, thereby improving the accuracy of the predicted binding pose. This allows for conformational changes during docking, as these residues are located within or near the binding site, increasing the likelihood of interaction with the ligand and influencing its binding pose. A total of 100 distinct conformations were generated and evaluated based on the best interaction energy with the enzyme and the best superposition parameters, considering both accommodation in the active site and conformations for the resurrection mechanism.

After parameter adjustment, the same procedure was carried out for the seven previously selected ligands, generating a total of 100 poses for each ligand, which were subsequently re-evaluated using the same criteria. Finally, the four best ligands were chosen.

### 2.2. QM/MM Procedure

The study of quantum mechanics in biomolecular complexes requires a high computational capacity due to the presence of a large number of atoms in these systems. In order to deal with certain rearrangements of covalent bonds that cannot be solely addressed by molecular mechanics, it was necessary to develop a hybrid technique of quantum mechanics and molecular mechanics (QM/MM). In this study, this approach was employed to examine the resurrection process of the enzyme–OP complex by the QMPs. The QM/MM approach selects a small portion of the system that is essential for the chemical reaction and treats it with quantum mechanics (QM), using methods such as *ab initio*, semi-empirical, or density functional theory (DFT). This strategy allows for the accurate assessment of electronic correlations in the region where bond breaking and formation occur. The rest of the system is treated with molecular mechanics (MM)-based methods. To determine the energy barrier for the reaction mechanism in the resurrection process of the aged AChE with the QMPs targeted in this study, QM/MM hybrid calculations were performed based on the selected poses from the docking procedure.

The QM/MM hybrid calculations in this study were performed using the Gaussian 09 program at the DFT level of theory and the 6-311 + G** basis set for the QM part. The active region was limited to a radius of 2.5 Å. The use of DFT methods is common in biomolecular studies due to their effectiveness in handling large systems.

All transition states (**TSs**), intermediates, and precursors involved were computed and characterized by calculating their imaginary frequencies. Each conformer underwent full optimization, in accordance with methodologies detailed in prior studies [1,27]. Reactants and products were linked through the potential energy curve. Each system was optimized using the B3LYP density functional, which utilizes the three-parameter exchange functional of Becke and the gradient-corrected functional of Lee, Yang, and Paar, along with the 6-311 + G** basis set, as described by da Cunha et al. (2008) [28]. Based on the obtained results, the most promising molecules were selected based on interaction energy and activation energy criteria for the most promising ligand. Additionally, following each optimization, a force constant calculation was conducted to confirm whether the optimized structures represented local minima (no imaginary frequencies) or transition states (one imaginary frequency).

### 2.3. Toxicity Testing and Absorption, Distribution, Metabolism, and Excretion (ADME) Testing

To evaluate the toxicity of the compounds using a safe computational method, an internet-based software, the ProTox-II 323 server, was employed to determine the toxic effects of the selected ligands based on their interaction energy, molecular docking overlay, and binding energy, as described by Banerjee and Ulker (2022) [29]. This platform is capable of measuring various toxicological endpoints, including the median lethal dose (LD50), immunotoxicity, cytotoxicity, and organ toxicity such as hepatotoxicity, as highlighted in previous studies [30,31].

### 2.4. ADME Testing (Absorption, Distribution, Metabolism, and Excretion)

ADME testing is an important assessment conducted on molecules with pharmacological potential [31]. The analysis of the solubility and pharmacokinetics of the key ligands proposed as drugs capable of resurrecting the aged AChE was performed using the Swiss-ADME server [32].

## 3. Results and Discussion

### 3.1. Docking Procedure

To validate the methodology and parameterize the calculations, redocking was performed to obtain the Root-Mean-Square Deviation (RMSD). RMSD is a commonly used measure to assess the accuracy of a molecular structure overlay in *in silico* studies, providing information about structural and positional differences [33]. The literature establishes that RMSD values below 2 Å are considered adequate for validating the redocking protocol [34].

In the redocking process, the enzyme with the crystallized ligand extracted from the PDB (PDB: **2WG1**) was subjected to molecular docking using the Molegro Virtual Docker (MVD) software. Among the 100 generated poses, the pose with the best interaction energy and conformation was selected and then overlaid onto the initial structure of the crystallized ligand. RMSD was calculated based on the difference in distance between the two structures [35]. Figure 3 shows the result of this overlay, with an RMSD value of 1.08 Å, validating the proposed protocol as it is below the 2 Å threshold presented in the literature [36,37].

The determination of the RMSD value not only validated the proposed protocol but also allowed for the definition of ideal parameters for other docking processes, such as the restriction radius, coordinates, and number of flexible amino acids.

After completing the validation and parameterization steps, 100 poses were obtained for each docked ligand, from which the interaction energies and hydrogen bond energies in kcal mol^−1^, and relative activities (experimental) in percentage were calculated, as presented in Table 1. The selection of the most promising ligands was based on the lowest interaction energy and the quality of the generated poses, as the affinity of the ligand to the receptor is assessed by predicting the preferred orientation and minimal interaction energy.

In Figure 4, the representation of the interaction results for the molecular docking process of ligands **C1**, **C2**, **C3**, **C4, C5,** and **C6** is presented.

In Figure 4, a 2D pharmacophore map is presented, highlighting the interactions between the ligands and the receptor. In the case of ligand **C1**, **π**-alkyl interactions can be observed between a pentane ring of the ligand and the residues Phe330 and Trp84. Additionally, the Trp84 residue forms a **π**-**π** interaction with the benzene ring of the molecule, along with other van der Waals interactions. On the other hand, ligand **C2** shows main interactions including a **π**-**π** interaction between the benzene ring and the Phe330 residue, a van der Waals interaction between the hydrogen of the benzene ring and the His440 residue, and two unfavorable interactions between the Glu199 residue and its aliphatic hydrogens. It is worth noting that the identification and characterization of these interactions are crucial to understand the affinity of the ligands with the receptor and guide the development of new molecules with higher potency and selectivity.

Ligands **C3** and **C4** exhibit similar interactions in molecular docking, such as Van der Waals, electrostatic, and hydrophobic interactions, which contribute to the stability of the inhibitor in the protein binding site. Two **π**-**π** interactions between the ligands and the His440 and Trp84 residues are highlighted, along with a hydrogen bond with the Glu199 residue. It is important to observe that ligands **C3** and **C4** interact with Glu199, while the interaction with His440 is observed in all ligands. The **C5** ligand interacts with the residues Phe330 and Trp84, where the Phe330 residue forms a **π**-**π** interaction with the benzene ring of the ligand. Its other interactions are primarily characterized by van der Waals forces. In contrast, the **C6** ligand predominantly exhibits pi–pi interactions, with the ring forming **π**-**π** interactions with the His440 and Trp84 residues. Additionally, Trp84 also participates in a **π**-alkyl interaction with one of the carbons of the ligand. Furthermore, the **C6** ligand presents an unfavorable interaction with the Glu199 residue, which may influence its affinity for the receptor. Thus, it is evident that the interactions with these amino acids are important features in determining the mode of interaction of the ligand with the OP, as the active ligand found in the redocking process also interacts with both of these amino acids.

By analyzing Table 1, it is noticeable that the intermolecular interaction energies in all ligands exhibited very close values, which was expected as a group of molecules from the same family was used, and the selection of their best poses in the docking study was based on the redocking calculations conducted at the beginning of this study. Furthermore, the hydrogen bonding interactions formed by these ligands, along with other contributing factors, increased their stability at the active site of the enzyme, and their structural parameters allowed for a good accommodation of the ligands in the active cavity, further enhancing their ligand–enzyme interaction affinity. It is important to highlight that interaction energies obtained solely from docking studies are insufficient to fully explain the experimental results, as evidenced by the low coefficient of determination (R^2^ = 0.67, Figure 5). This coefficient was determined by correlating the ΔE values obtained in the docking study with the experimental values of relative % resurrection. The low coefficient of determination suggests that additional factors beyond interaction energy play a significant role in the resurrection process. Therefore, considering additional parameters is crucial to enhance the accuracy of the theoretical–experimental correlation in QM/MM calculations. For this reason, theoretical analyses at the quantum level were conducted to determine the relative resurrection energy of the ligands.

### 3.2. Mechanism Study

Understanding the mechanism behind the resurrection process of the acetylcholinesterase (AChE) enzyme is an essential step in the development of new, more selective and effective ligands [38]. It is important to consider the dynamic effects on the reaction mechanism and the orientation of the ligand to gain a complete understanding of the ligand–AChE interaction [39,40]. In this QM/MM study, we compared the ΔE* values between the transition state and the initial system for each system (Table 2). In this way, we obtained the QMP’s reactivity tendency, thus avoiding the direct computation of the absolute energy values.

After optimization of the selected compounds, an infrared calculation was performed to ensure that the structures reported in Table 2 are all transition states. From the transition states, the reaction paths were traced to obtain the geometries of lowest energy.

Table 2 presents the kinetic parameters, predicted **ΔΔE^#^** values from the theoretical calculations of the ligands, and the corresponding experimental values to the respective imaginary frequencies which characterize the **TS** structures. The kinetic parameters **ΔE*** and ΔΔE# are intrinsically related to the rate constants of a chemical reaction through the Arrhenius equation. When examining transition state (**TS**) structures, it is noted that chemical and structural changes in ligands play a crucial role in stabilizing the enzyme. Our primary focus is on accurately identifying saddle points on the potential energy surface. This process involves frequency analysis to confirm the nature of the transition states, where a single imaginary vibrational mode corresponds to the principal reaction coordinate. Furthermore, characterizing the eigenvectors associated with the vibrational frequencies of the transition states provides crucial insights into atomic movements during the reaction process. As observed earlier, the interaction energies from the docking calculations show very similar values among the studied compounds, which is why the experimental data cannot be adequately explained solely based on the interaction energy values. Additionally, regarding the reaction between the ligand and the neurotoxic agent, the ΔΔE# values were found to be inadequate in explaining the experimental results (R^2^ = 0.0015, Figure 6). The low coefficient of determination observed in Figure 6 indicates a very weak correlation between the resurrection percentage and the ΔΔE# values, suggesting that these kinetic parameters are insufficient to fully elucidate the experimental results of resurrection. Therefore, there is an evident need to consider other relevant effects associated with the binding process in order to achieve a more accurate correlation between theory and experiment.

It is evident that the process of AChE resurrection depends on two fundamental steps: the association of the ligand with the aged enzyme (AChE-LIG) and the chemical reaction between the ligand and the neurotoxic agent (OP-LIG) [38]. Based on the data presented in Table 2, a mathematical model was developed using the multiple linear regression (MLR) method, resulting in the formulation of Equation (1) for predicting the percentage of resurrection in the investigated systems.
(1)$$\%Resurrection=-0.835\left(∆E\right)-2.143\left(∆∆E\#\right)-70.536$$

According to the theoretical results presented in Equation (1), the association of the ligand with the aged enzyme and the subsequent chemical reaction are essential factors that significantly influence the AChE resurrection process. The equation developed to predict the resurrection percentage of the investigated systems showed a good correlation (R^2^ = 0.75) between the theoretical (ΔE and ΔΔE#) and experimental values of relative % resurrection, highlighting the importance of these steps for successful reactivation. The data used to find this correlation can be found in Table 2. Reduced ΔΔE# values correspond to a significant increase in % resurrection, while lower binding energy values result in a higher % resurrection. It is worth noting that although ligand **C2** has a lower ΔΔE# value, its experimental % resurrection is very small. However, ligand **C4** presents good interaction energy and a low ΔΔE# value, showing a promising experimental trend. Additionally, this ligand is similar to ligand C8 in the study by Zhuang et al. (2018) [23], reinforcing its efficacy both theoretically and experimentally. Based on these results, new promising molecules are proposed based on the interaction values and resurrection energy of the most promising ligand.

### 3.3. Design of New Ligands

Based on the results of the previous study, new promising molecules were designed guided by the in silico and in vitro results of the most effective ligand. Structural modifications were made to ligand **C4** (Figure 7), resulting in new compounds (Figure 8), and a representation of the interaction results for the molecular docking process of ligands **C4-1**, **C4-2**, **C4-3**, **C4-4**, **C4-5**, and **C4-6** is presented in Figure 9. It is worth noting that the conformation of ligand **C4** in the active site of aged AChE occupies a large space in the cavity, suggesting that structural modifications could disrupt the interaction between the ligand and the active site. Another aspect to consider in the modifications of ligand **C4** is its degree of hydrophobicity, which can significantly influence both its interaction with the enzyme and its resurrection capability. Hydrophobic ligands tend to bind more strongly to hydrophobic regions of the enzyme, increasing the stability of the binding and the ligand’s affinity for the active site. Moreover, hydrophobicity can impact the solubility of the ligand and its distribution in the biological environment, affecting its efficacy and specificity. The ligand’s hydrophobicity facilitates stronger van der Waals interactions with hydrophobic enzyme residues, which significantly contributes to the stability of the ligand–enzyme complex. Therefore, the search for an ideal ligand requires balancing hydrophobicity and polarity, also considering its affinity for the enzyme, resurrection capability, and potential toxicity during structural modifications. Considering that the critical step in the resurrection of acetylcholinesterase is the binding between the ligand and the organophosphorus (OP) compound, the proposed structural modifications for the new ligands aimed to stabilize the leaving group by adding resonance from the pentane ring and/or adding nitrogen at different positions in the ring, as well as adding hydroxyl groups to the pyridine ring, to improve the interaction with the AChE active site and optimize the interaction energies.

The discrepancies among ligands **C4-1, C4-2, C4-3**, and **C4-4,** regarding the insertion of double bonds and the addition of nitrogens in the pentane ring, exert a significant impact on their properties and interactions with the active site of aged acetylcholinesterase. The main alterations in the structures were in the insertion of double bonds and nitrogen atoms in the ring. Both structures **C4-1** and **C4-4** have only one nitrogen, both located at position 1 of the ring; however, **C4-1** features two additional double bonds, whereas **C4-4** has only one. On the other hand, structures **C4-2** and **C4-3** exhibit two double bonds and two nitrogen atoms. The crucial difference between **C4-2** and **C4-3** lies in the position of the nitrogens: in **C4-2**, they occupy positions 1 and 3, whereas in **C4-3**, they are at positions 1 and 2. These double bonds help stabilize the ligand’s conformation and maximize **π-π** interactions with aromatic residues in the active site, while the addition of nitrogens increases the potential for hydrogen bond formation. Additionally, the presence of nitrogens can alter the electronic distribution of the ring, affecting binding affinity and the stability of interactions with the active site. For ligands **C4-5** and **C4-6**, the **OH** group was added in different positions on the ring. This group was included to act as a hydrogen bond donor or acceptor, allowing for favorable interactions with charged residues in the active site. The addition of the **OH** group also aims to increase the ligand’s affinity and specificity. The presence of the **OH** group can restrict the ligand’s flexibility, limiting its possible conformations in the active site. If the new conformation is favorable, the affinity will be increased. These subtle structural modifications have a direct impact on the electronic and geometric properties of the ligands, being crucial for optimizing their effectiveness in the process of acetylcholinesterase resurrection.

Six distinct ligands were constructed based on the structure of molecule **C4**, following the modifications carried out to its structure. These ligands predominantly exhibited interactions with the residues Trp84 and Glu199. Ligands **C4-1** and **C4-2** showed, in addition to these common interactions, **π**-**π** interactions with the residues Gly117, His440, and Phe330. On the other hand, ligands **C4-3** and **C4-4** displayed notable **π**-**π** interactions with the residues Trp84 and Phe331, although ligand **C4-4** exhibited an unfavorable interaction with the amino acid Glu199. The ligand **C4-5** exhibited **π**-**π** interactions with the residues Phe330 and Trp84. Additionally, it formed two **π**-alkyl interactions with these same residues. However, unfavorable interactions were observed with the residue Phe330. On the other hand, the ligand **C4-6** demonstrated a **π**-alkyl interaction with the residue Trp84 and hydrogen bonds with the residues Ser200, Glu199, and Mol604. It is important to note that the identification and characterization of these interactions are crucial to understanding the affinity of the ligands for the receptor and guiding the development of new molecules with increased potency and selectivity. It is noteworthy that the interaction with residues His440 and Glu199 is an important aspect in determining the mode of interaction of the ligand with the OP, as the active ligand found in the redocking process exhibits interactions with both amino acids.

By analyzing the energy profiles of the conformations of ligands **C4-1**, **C4-2**, **C4-3**, **C4-4, C4-5,** and **C4-6** compared to the original **C4** ligand, it can be observed that ligands **C4-2** and **C4-3** exhibited lower interaction energies than those obtained in the docking of the **C4** ligand. In contrast, ligands **C4-1** and **C4-4** showed higher energies compared to the original ligand, but their interaction energies were lower than the values found in ligands **C1**, **C2**, and **C3,** as analyzed in the first part of the study (Table 2). This confirms that similar structures to ligand **C4** have a better interaction with the enzyme.

Furthermore, upon analyzing the differences in binding energy found in the ligand–enzyme complex in an intermediate energy state, it can be observed that ligand **C4** exhibited a smaller energy variation compared to the other ligands derived from its structure. However, to evaluate if these ligands have the ability to resurrect AChE activity, it is necessary to analyze the resurrection percentage (**% Resurrection**) of these ligands and compare it with that of ligand **C4** [41]. For this purpose, the values of intermolecular energy and binding energy were inserted into Equation (1), and the results are presented in Table 3.

Based on the presented results, it was observed that the ligands with lower binding energy and intermolecular interaction values with the enzyme were those that exhibited higher resurrection percentages, which shows agreement with Equation (1). Ligand **C4-2** showed a higher resurrection percentage compared to ligand **C4**, suggesting greater efficacy in relation to the latter. Additionally, ligand **C4-3** showed promise by displaying an excellent resurrection rate and can be considered capable of resurrecting AChE, similar to ligand **C4**. This result reinforces the idea that for an agent to be able to resurrect AChE, it needs to adopt a position that allows for attack on the phosphorus atom of the OP. This position is favorable for ligands based on the **C4** structure.

### 3.4. Toxicity Testing

In order to investigate the possibility of considering future clinical trials for a compound, it is crucial to evaluate its toxicity. The identification of parameters related to toxicity is a basic requirement for a drug candidate to be considered for the treatment of diseases [31,42]. The toxicity test results for compounds **C4** and the newly proposed ligands were conducted using the ProToxII platform [42] and are presented in Table 4. These compounds were identified as the most promising for future testing.

The toxicity of ligands **C4, C4-5,** and **C4-6** was classified at level 3, indicating no toxicity in any of the analyzed requirements, and their oral lethal dose (**LD50**) was 195 mg/kg. The ligands **C4-1**, **C4-2**, and **C4-3** also showed no toxicity, but their **LD50** ranged from 650 mg/kg to 837 mg/kg, classifying them at level 4 on the toxicity scale. On the other hand, **C4-4** had a lethal dose of 1600 mg/kg, but it exhibited hepatotoxicity, resulting in a level 4 classification on the scale of toxic compounds. Compounds that show toxicity in any tissue type, along with a low **LD50**, are generally considered harmful if ingested. From this perspective, an ADME test needs to be conducted to assess if the compound is promising in regard to becoming a drug [42].

In order to obtain information about the physicochemical properties of potential oral drug candidates, the Swiss-ADME platform was employed to provide parameters such as lipophilicity (WLOGP and TPSA), water solubility (ESOL log S), drug similarity rules, and medicinal chemistry [43]. By using a combination of Lipinski, Veber, Ghose, Egan, and Muegge rules, the ADME prediction study provided information on these properties. The results are presented in Table 5.

According to the ADME results, the potential drug candidates analyzed have molecular masses lower than the range required by the Muegge rule. However, the compounds met the other proposed rules for the number of rotational bonds, hydrogen acceptors, and donors. Regarding solubility, the compounds were evaluated based on log S values, with higher values indicating greater solubility. Additionally, TPSA and WLOGP values were assessed, as well as gastrointestinal absorption capacity, and both ligands comply with the rules. In the context of drugs, it is important to highlight that water solubility and lipophilicity are crucial characteristics for the effectiveness of a compound. In this regard, it is worth noting that among the evaluated ligands, ligand **C4-4** is the most hydrophobic and therefore the most water-insoluble, while ligand **C4-1** is the most hydrophilic. The permeability coefficient (kp) is an important measure in determining the skin permeation of a compound. The logarithmic transformation of kp, known as log kp, is often used to assess the ability of the compound to penetrate the skin. According to the literature, the more negative the log kp value, the lower the likelihood of the compound being absorbed through the skin, indicating low skin permeability [44,45]. In this case, the skin permeability results indicated that ligands **C4** and **C4-1** have the highest kp values, which are −6.78 and −6.74, respectively.

Lastly, the results of the ADME analysis indicate that none of the compounds deviated from the suggested standards for a molecule to be considered a drug. The compounds violated only one of the rules proposed by Lipinski, Veber, Ghose, Egan, and Muegge, indicating that they are viable options for an oral drug.

The results of this study revealed a significant difference in the resurrection efficacy between ligand **C4-2** and ligand **C4**, highlighting the potential of **C4-2** as a more efficient agent in reversing AChE aging. These findings indicate that **C4-2** is a promising candidate for future studies and the development of resurrection agents. Additionally, ligand **C4-3** also showed promising results, exhibiting a satisfactory resurrection rate, suggesting its potential in restoring AChE activity, similar to ligand **C4**.

It is important to highlight that ligands **C4-2** and **C4-3** demonstrated low toxicity, including lower **LD50** values compared to ligand **C4**, which is currently considered the best known compound for AChE resurrection. Furthermore, these ligands complied with the screening principles proposed by Lipinski, Veber, Ghose, Egan, and Muegge in ADME, strengthening their viability as effective in silico resurrection agents.

These results suggest that compounds **C4-2** and **C4-3** (Figure 10) are highly promising in resurrecting aged AChE and warrant further scientific investigations. Additionally, the favorable characteristics exhibited by these ligands, such as low toxicity and compliance with screening principles, make them ideal targets for future studies involving organic synthesis and biological evaluation.

## 4. Conclusions

In this study, computational chemistry methods were employed to assist in the process of discovering and identifying new ligands capable of resurrecting AChE, which remains a mystery in science. It is important to note that additional experimental investigations should be conducted to validate the theoretical results presented here. The obtained results suggest that the theoretical methodology employed in this work is suitable for the analysis of the studied systems. Indeed, the theoretical approach was able to qualitatively predict the affinity and reactivity between the ligands and AChE, and mathematically predict the **% resurrection** achieved by each ligand. Considering this, and being aware that the presented results are just the beginning of an arduous effort that requires further testing and studies, this work reports that new compounds based on QMP structures exhibited promising efficacy in resurrecting AChE aged by soman. **C4-2** and **C4-3** could be used as a starting point for subsequent studies involving organic synthesis and biological evaluation.

## Figures and Tables

**Figure 1 molecules-29-03684-f001:**
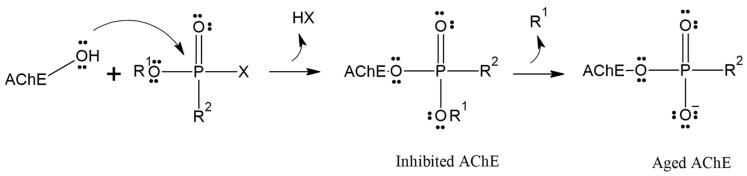
Inhibition and aging of acetylcholinesterase.

**Figure 2 molecules-29-03684-f002:**
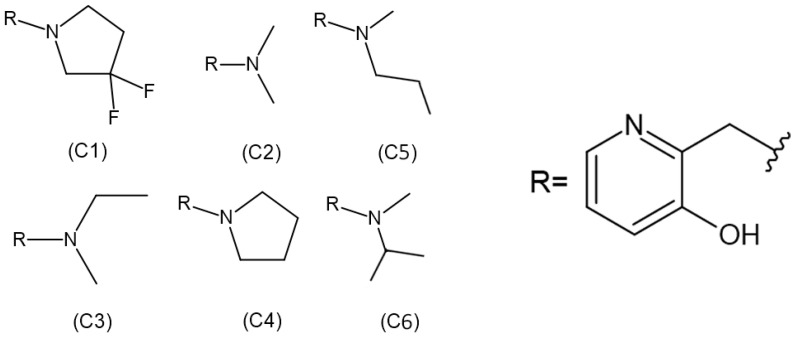
Structures of realkylator candidates.

**Figure 3 molecules-29-03684-f003:**
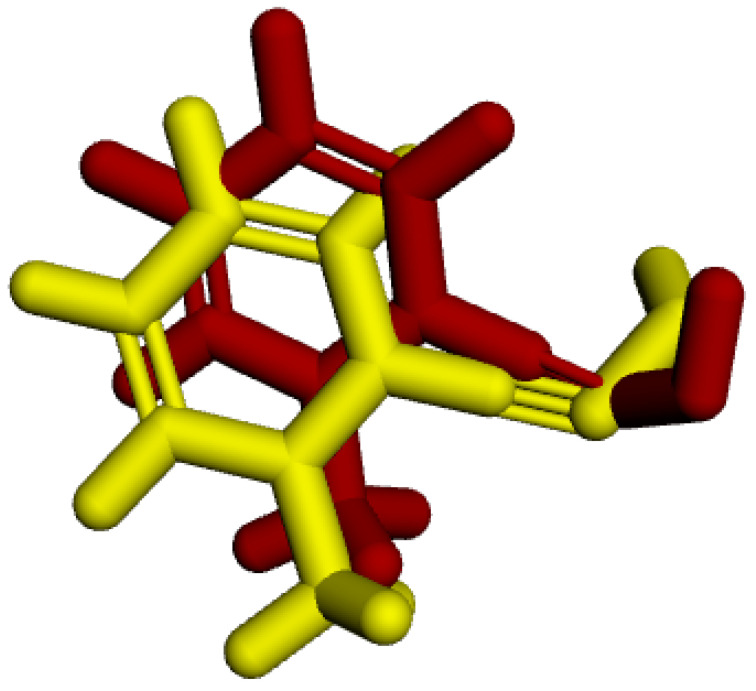
RMSD of the crystallized ligand LA (yellow) and the best redocking position (red).

**Figure 4 molecules-29-03684-f004:**
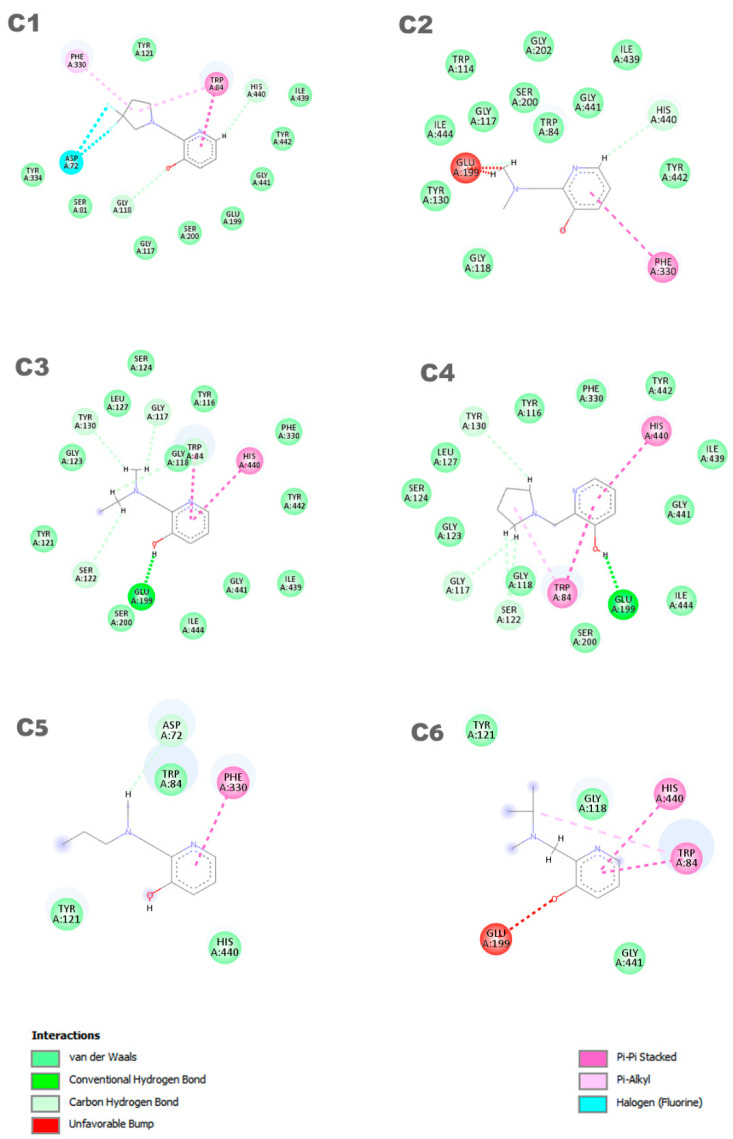
Interactions between the aged AChE enzyme and the ligands.

**Figure 5 molecules-29-03684-f005:**
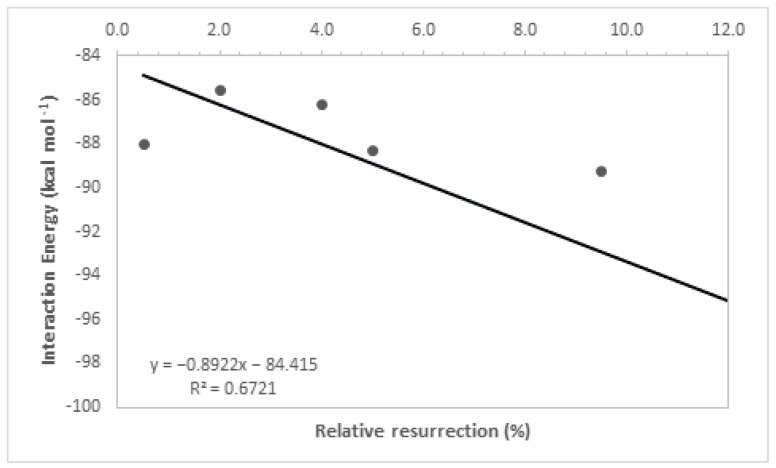
Resurrection percentage versus interaction energy graph.

**Figure 6 molecules-29-03684-f006:**
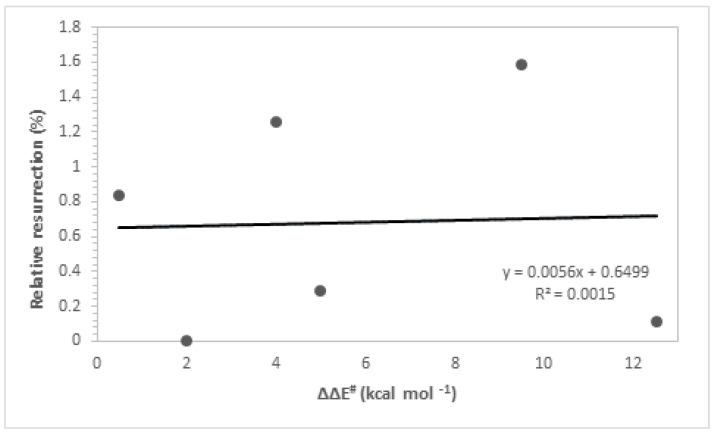
Resurrection % versus ΔΔE^#^.

**Figure 7 molecules-29-03684-f007:**
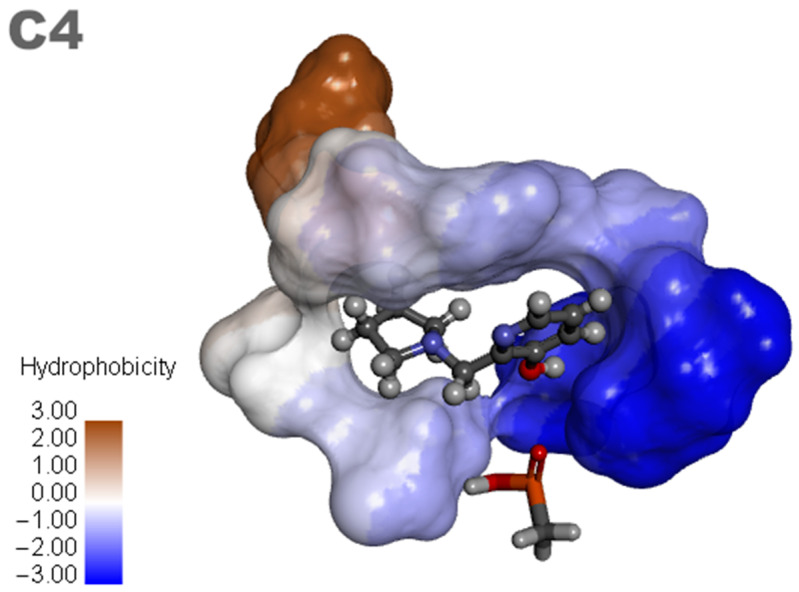
Ligand **C4** and analogues.

**Figure 8 molecules-29-03684-f008:**
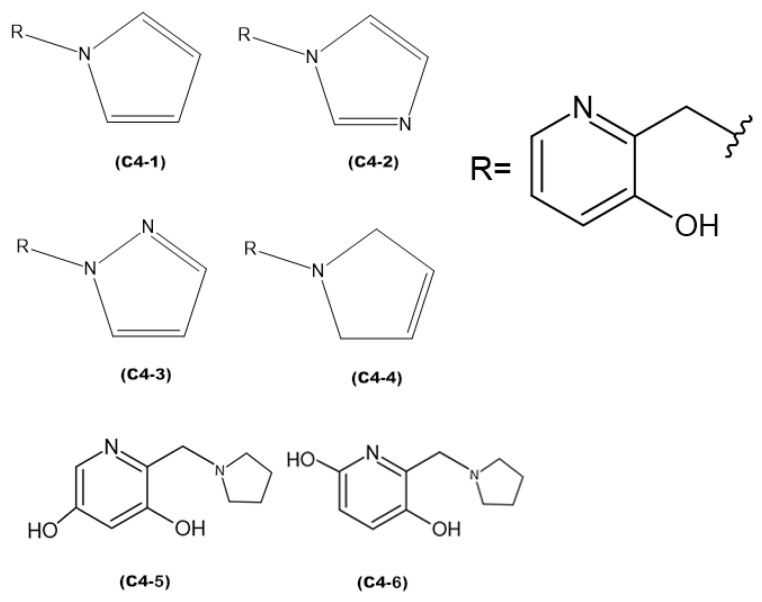
Proposed new ligands.

**Figure 9 molecules-29-03684-f009:**
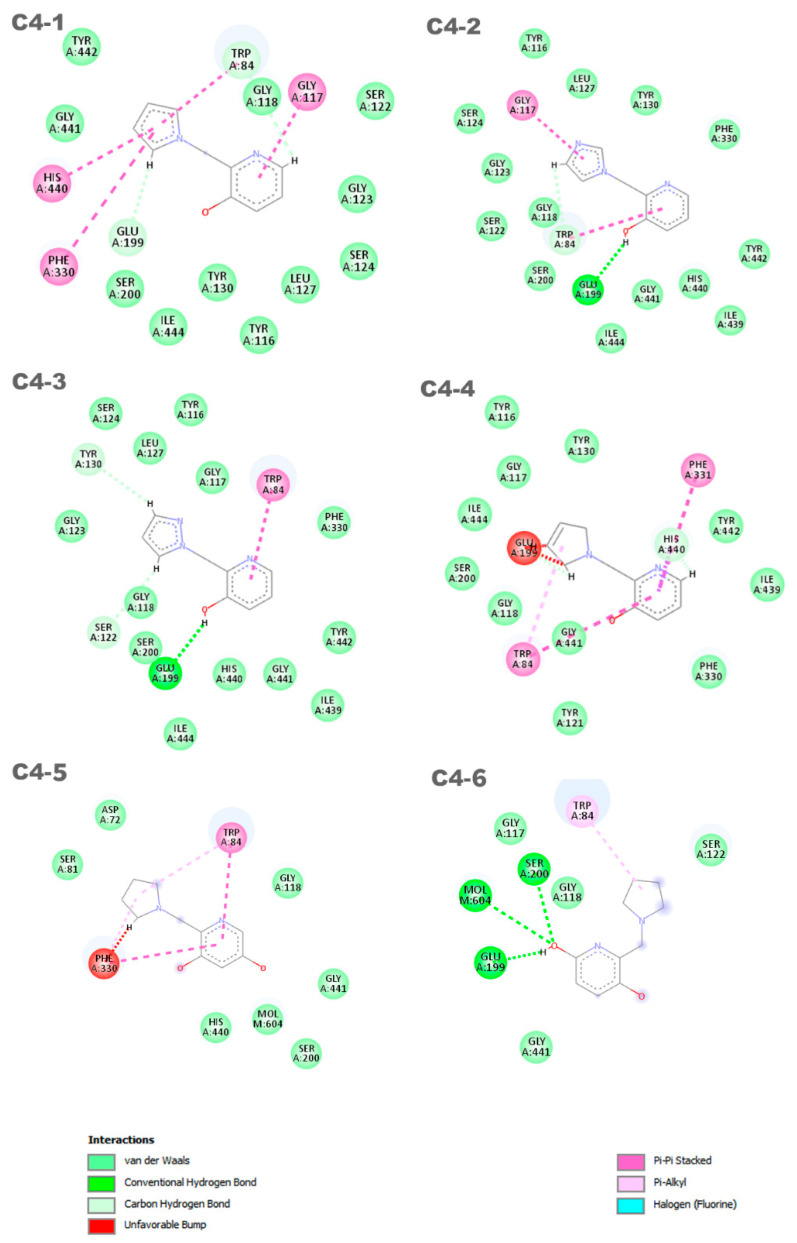
Interactions between the aged AChE enzyme and the proposed new ligands.

**Figure 10 molecules-29-03684-f010:**
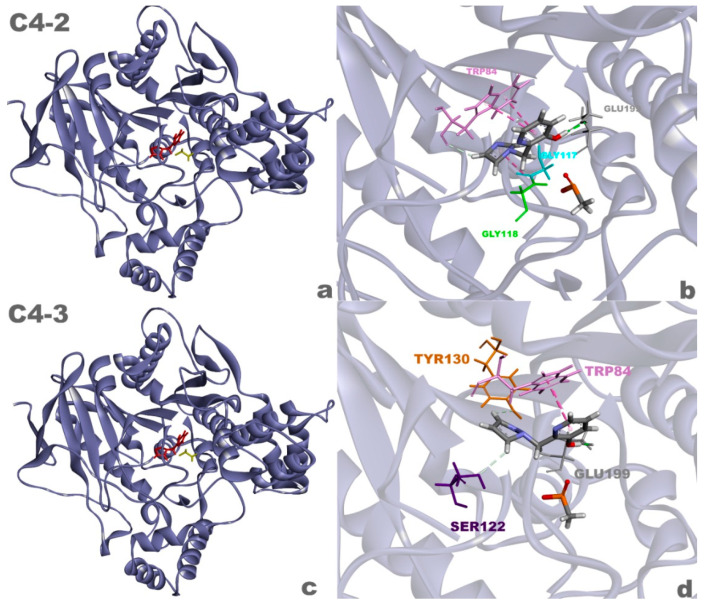
Complexes of the aged AChE enzyme with ligands **C4-2** (**a**) and **C4-3** (**c**), and 3D interactions between ligands **C4-2** (**b**) and **C4-3** (**d**) and the enzyme.

**Table 1 molecules-29-03684-t001:** Docking results and % reactivation values for OP-aged AChE realkylator candidates (**C1**–**C6**).

	Aged AchE		
	ΔE * (kcal mol^−1^)	*Relative Resurrection* ** *(*%*)*	H-Bond Energies (kcal mol^−1^)
**Active ligand**	−91.81	*-*	−5.00
**C1**	−88.00	*0.50*	−4.99
**C2**	−85.55	*2.00*	0.00
**C3**	−89.18	*9.50*	−2.50
**C4**	−99.18	*12.50*	−2.50
**C5**	−86.19	*4.00*	−2.49
**C6**	−88.27	*5.00*	−2.50

ΔE * = Intermolecular energy; ** 4 mm of QMP [23].

**Table 2 molecules-29-03684-t002:** Experimental results, intermolecular energy, and relative activation energies.

Ligand	*Experimental (Relative Resurrection* %*)*	ΔE * (kcal mol^−1^)	ΔΔE ^#b^ (kcal mol^−1^)	Frequency/cm^−1^
**C1**	*0.50*	−88.00	0.84	i81.95
**C2**	*2.00*	−85.55	0.00	i109.60
**C3**	*9.50*	−89.18	1.59	i180.50
**C4**	*12.50*	−99.18	0.11	i89.77
**C5**	*4.00*	−86.19	1.25	i92.88
**C6**	*5.00*	−88.27	0.29	i95.75

* ΔE = Intermolecular energy. ^b^ΔΔE^#^ = Δ_LIG2_ − Δ_LIG1._

**Table 3 molecules-29-03684-t003:** Results of intermolecular energy, relative activation energies, and resurrection percentage.

Ligand	ΔE kcal mol^−1^	ΔΔE (kcal mol^−1^)	% Resurrection	Frequency/cm^−1^
**C4**	−99.18	0.114	12.03	i89.77
**C4-1**	−98.84	2.540	6.55	i149.08
**C4-2**	−100.73	0.574	12.34	i126.96
**C4-3**	−101.56	1.003	12.11	i177.40
**C4-4**	−95.37	1.908	5.00	i140.24
**C4-5**	−81.962	0.099	−2.31	i102.14
**C4-6**	−85.291	0.115	0.43	i108.61

**Table 4 molecules-29-03684-t004:** Toxicity testing results.

Endpoint	Target	C4	C4-1	C4-2	C4-3	C4-4	C4-5	C4-6
Organ Toxicity	Hepatotoxicity	Inactive	Inactive	Inactive	Inactive	Active	Inactive	Inactive
Toxicity EndPoints	Carcinogenicity	Inactive	Inactive	Inactive	Inactive	Inactive	Inactive	Inactive
	Immunotoxicity	Inactive	Inactive	Inactive	Inactive	Inactive	Inactive	Inactive
	Mutagenicity	Inactive	Inactive	Inactive	Inactive	Inactive	Inactive	Inactive
	Cytotoxicity	Inactive	Inactive	Inactive	Inactive	Inactive	Inactive	Inactive
	LD50 (mg/kg)	195 mg/kg	800 mg/kg	837 mg/kg	650 mg/kg	1600 mg/kg	195 mg/kg	195 mg/kg
	Toxicity	3	4	4	4	4	3	3
Tox21—Nuclear	Aryl hydrocarbon Receptor (AhR) Androgen Receptor (AR)	Inactive	Inactive	Inactive	Inactive	Inactive	Inactive	Inactive
receptor signalling								
pathways								
Tox21—Stress response	Heat shock factor	Inactive	Inactive	Inactive	Inactive	Inactive	Inactive	Inactive
pathways	response element							
	(HSE)							

**Table 5 molecules-29-03684-t005:** ADME results.

Molecule	MW (g/mol)	Rotable Bonds	H-Bonds Aceptors	H-Bonds Donors	ESOL Log S	TPSA(Å^2^)	WLOGP	GI Absorption	Log Kp (cm/s)
**C4**	178.23	2	3	1	−1.70	36.36	0.82	**High**	−6.78
**C4-1**	174.20	2	2	1	−1.90	38.05	1.64	**High**	−6.74
**C4-2**	175.19	2	3	1	−1.60	50.94	1.03	**High**	−7.21
**C4-3**	175.19	2	3	1	−1.70	50.94	1.03	**High**	−7.06
**C4-4**	176.22	2	3	1	−1.50	36.36	0.63	**High**	−6.95
**C4-5**	194.23	2	4	2	−1.5	56.59	0.56	**High**	−7.13
**C4-6**	194.23	2	4	2	−1.7	56.59	0.56	**High**	−6.90

## Data Availability

Data is contained within the article.

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
