# Peer review of "Atomistic Origins of Resurrection of Aged Acetylcholinesterase by Quinone Methide Precursors"

_molecules, 2024, doi:10.3390/molecules29153684_

Round 1

Reviewer 1 Report

Comments and Suggestions for Authors

Report molecules-3010947

Authors perform a computational study of the resurrection of aged acetylcholinesterase by quinone methide precursors. The main goal is to design a set of new ligands with the objective of improve the resurrection percentage. The question to be solved is scientific relevant. Nevertheless, major issues regarding the methodology should be solved before publication. The actual version of the manuscript displays some ambiguities regarding the concept of transition state and transition state energies, that are relevant to justify the conclusions.

Major points

1.     Authors claim to perform a mechanistic study. Nevertheless, the procedure to the calculation of transition states is not provided. In addition, no information of such transition states is discussed. The following questions  should be discussed in the text of the manuscript :

a.     What structural-chemical change is related with this transition states?

b.     Are those transition states characterized using a frequency analysis?

c.     are the eigenvectors related to the connection between reactants and products relevant for the characterization of the mechanism?

2.     Which kind of kinetic parameters are related with DE or DDE#? Are those values related with kinetic constants through Arrhenius equation? Please clarify such points.

3.     Regarding the % Resurrection linear adjustment, I think that authors need to use a larger data to conclude that a strong correlation is obtained, and to use such equation to calculate %Resurrection for new ligands.

4.     Regarding the new ligands purposed, please justify the molecular modifications using a pharmacophoric model or an extensive discussion of the needed properties of the ligand based on the characteristic of the active site more than hydrophobicity, hydrogen bond interactions, pi-pi stacking, etc.

5.     In page 12 authors write:

“Furthermore, upon analyzing the differences in binding energy found in the 377 transition state of the ligand-enzyme complex, it can be observed that ligand C4 378 exhibited a smaller energy variation compared to the other ligands derived from its 379 structure”

What means a “transition state of the ligand-enzyme complex”? Seems that authors misunderstand the concept of transition state (https://doi.org/10.1021/jp953748q).

Author Response

Dear Editor,

          Please find enclosed the corrected version of the manuscript “Atomistic Origins of Resurrection of Aged Acetylcholinesterase by quinone methide precursors” by Ferreira et al.

The authors explored Atomistic Origins of Resurrection of Aged Acetylcholinesterase by quinone methide precursors. The aim of the paper is meaningful and interesting. However, the authors should explain better the terms in the introduction and in the methodology to improve the understanding of the manuscript. In addition, the novelty of the paper should be well highlighted in the introduction.

Some specific commentaries are presented bellow:

Introduction

The authors should be more objective in the first four paragraphs of the introduction. The authors should concentrate on explaining better the action of irreversible inhibition of ACHE and its toxic effects. Adding a figure in the introduction can help to explain better this phenomenon.

  • R: Thank you for this valuable comment. In response to the reviewer's suggestion, we have added a new figure. Additionally, new paragraphs have been included in the Introduction section.

Explain better the difference between the aging of AChE and inhibition of AChE.

  • R: We acknowledge this important observation. In the introduction section of the manuscript, starting from the eighth paragraph, the difference between inhibited and aged AChE was explained. Additionally, a figure illustrating the difference between the two forms has been added.

Explain the function of molecular docking analysis and QM/MM Procedure to achieve the main objective of the paper.

  • R: Thank you for this valuable comment. In the eleventh paragraph of the introduction, the revised manuscript text has been updated to include the role of molecular docking analysis and the QM/MM procedure conducted in our study and their objectives. These methods were used to investigate the resurrection of QMPs in enzyme-OP complexes. By using molecular docking poses and QM/MM calculations, our results were able to determine the resurrection energy barrier for each enzyme-OP complex with the QMPs. This not only validates the effectiveness of the QM/MM approach but also provides detailed insights into the reaction pathway and the energy barriers involved.

Methodology

Figure 1: The legends of figures should be auto-explained. Therefore, put the meaning of c1 – c4.

  • R: The legends of Figure 2 has been modified in the revised manuscript to make it more auto-explained, and this figure has been updated to include two additional ligands in the subsequent calculations.

New figure 2.

In the phrase: “The ligand 534_FP1065 (LA), present in the crystal structure 145 of 2WG1, was subjected to the re-docking process with a spherical restraint of 10 Å, 146 where the amino acids were considered flexible”. Please, explain better this part of the method. Which criteria were used to consider the amino acid flexible?

  • R: Thank you for this observation. Now, we have corrected this commented sentence. In fact, the ligand-enzyme system was considered flexible, as flexible docking was used; all species in the complex are kept free, allowing them to adjust their positions to optimize interactions, thereby improving the accuracy of the predicted binding pose. This sentence has been corrected in the revised manuscript. The parameters used to obtain the docking structures included atomic volumes and cavities, as well as the charges associated with the atoms involved. The ligand 534_FP1065 (LA) and the other ligands were subjected to docking calculations with a spherical constraint of 10 Å. The scoring function used was Moldock score [GRID] with a resolution of 0.3 Å, and the algorithm employed was MolDock Optimizer with 100 runs, restricted to 9 Å, and residues within 12 Å of the 79.86 cm³ cavity in the enzyme's active site. The ligand-enzyme system was considered flexible for all calculations.

Results

Table 1. The legends of tables should be auto-explained. Therefore, put the meaning of c1 – c4 and âˆ†E. How was the relative activity calculated?

  • R: Thank you for this observation. In this version, the legend of Table 1 has been modified in the final manuscript to make its caption more self-explanatory. There was an error in one of the terms cited in this table: where it said relative activity, it should have been relative resurrection. This correction has already been made in the revised manuscript. Regarding the question, the relative resurrection was found based on the experimental work conducted by Zhuang and colleagues in 2018.

Table 2: How was the experimental relative resurrection calculated? put the meaning of âˆ†E and âˆ†âˆ†E.

  • R: Thank you for this observation. The relative resurrection was determined from the experimental work conducted by Zhuang and colleagues in 2018. In their study, they performed an experimental screening where aged AChE reacted with different concentrations (0, 2-4 mM) of various QMPs for one day. As a result, they presented the relative resurrection values for each ligand tested at these different concentrations. The terms ΔE and ΔΔE# represent the activation energy and ΔΔE# = ΔLig2 - ΔLig1, which indicates the reactivity trend of the QMPs, thereby avoiding the direct calculation of absolute energy values. These terms are essential for understanding the reaction kinetics. The use of these terms were clarified in the revised version of the manuscript.

The authors are encouraged to perform an analysis of toxicity on experimental systems.

  • R: We agree with the reviewer’s comment. However, we believe that our theoretical information is fundamental for future work. Even though this is a theoretical study, our research is based on an analysis of the interactions between the enzyme's active site and the ligands and their % resurrection. We address the relevant molecular interactions, including, but not limited to, hydrogen bonding, hydrophobic interactions, van der Waals interactions, and π-π Additionally, we conducted theoretical ADME and toxicity studies. These computational techniques provide valuable information about the pharmacokinetic and toxicological properties of the compounds. Our pharmacophore, ADME, and theoretical toxicity studies were crucial for ligand optimization, ensuring that the molecular modifications resulted in compounds with favorable pharmacokinetic and toxicological profiles. We believe that the combination of active site interaction analysis with pharmacophore, ADME, and theoretical toxicity studies, although still under development, provides a good starting point for justifying the molecular modifications made to the new ligands.

The legend of Table 5 should be auto-explained.

  • R: Thank you for this comment. The legend for Table 5 has been modified to make it more auto-explanatory.

It is important to validate at least the best results obtained using experiments

  • R : We agree with the reviewer comment. In line with that, we believe that our theoretical information will be essential in any future effort towards design and discovery of quinone methide precursors. In this context, theoretical tools can be employed as filters to flag and de-select the potentially harmful compounds at the preclinical stage of drug development, thereby potentially avoiding significant economic and human health consequences incurred at later stages of drug discovery. To date, ongoing research efforts to develop new compounds for resurrection of AChE aged by the OP Soman based on new structures of quinone methide precursors are important and work along this line is in progress. Thus, we believe to report our current theoretical findings in the literature can stimulate new experiments and other theoretical investigations that could assess the validity of this assumption.

We acknowledge again the referee’s comments, which have enabled us to significantly improve our paper.

Furthermore, we thank the editorial assistance and hope with the changes and clarifications implemented, the manuscript would be now acceptable for publication in Molecules. Finally, we also remain at your disposal for any further inquiries.

With best regards,

Authors

Reviewer 2 Report

Comments and Suggestions for Authors

The authors explored Atomistic Origins of Resurrection of Aged Acetylcholinesterase by quinone methide precursors. The aim of the paper is meaningful and interesting. However, the authors should explain better the terms in the introduction and in the methodology to improve the understanding of the manuscript. In addition, the novelty of the paper should be well highlighted in the introduction.

Some specific commentaries are presented bellow:

Introduction

The authors should be more objective in the first four paragraphs of the introduction. The authors should concentrate on explaining better the action of irreversible inhibition of ACHE and its toxic effects. Adding a figure in the introduction can help to explain better this phenomenon.

Explain better the difference between the aging of AChE and inhibition of AChE.

Explain the function of molecular docking analysis and QM/MM Procedure to achieve the main objective of the paper.

Methodology

Figure 1: The legends of figures should be auto-explained. Therefore, put the meaning of c1 – c4.

In the phrase: “The ligand 534_FP1065 (LA), present in the crystal structure 145 of 2WG1, was subjected to the re-docking process with a spherical restraint of 10 Å, 146 where the amino acids were considered flexible”. Please, explain better this part of the method. Which criteria were used to consider the amino acid flexible?

Results

Table 1. The legends of tables should be auto-explained. Therefore, put the meaning of c1 – c4 and ∆E. How was the relative activity calculated?

Table 2: How was the experimental relative resurrection calculated? put the meaning of ∆E and ∆∆E

The authors are encouraged to perform an analysis of toxicity on experimental systems.

The legend of Table 5 should be auto-explained.

It is important to validate at least the best results obtained using experiments

Author Response

Dear Editor,

We greatly appreciate your detailed and insightful comments on our manuscript. We recognize the importance of the issues raised and are fully committed to addressing them comprehensively and satisfactorily.

  1. Authors claim to perform a mechanistic study. Nevertheless, the procedure to the calculation of transition states is not provided. In addition, no information of such transition states is discussed. The following questions should be discussed in the text of the manuscript :

  1. What structural-chemical change is related with this transition states?

  • R1(a.): Thank you for this important comment. The chemical-structural changes in the ligands are directly related to the stabilization of the transition states through interactions made between the ligand and the active site of the enzyme. These interactions are key factors that determine these properties and, consequently, the efficacy of the ligands in acetylcholinesterase resurrection processes. In this new version, we have included a short sentence in order to discuss this part.

  1. Are those transition states characterized using a frequency analysis?

  • R1(b.): Thank you for your comment. In this revised version, we have incorporated theoretical results from frequency analysis. In fact, our transition state analysis concentrates on identifying and characterizing saddle points on the potential energy surface using computational methods rooted in density functional theory (DFT). Frequency analysis is performed to confirm the nature of the saddle point, as indicated by a single imaginary mode corresponding to the reaction coordinate.

  1. are the eigenvectors related to the connection between reactants and products relevant for the characterization of the mechanism?
  • R1(c.): In fact, we agree with the reviewer and in line with his/her comment, we have included a short sentence in the manuscript. The eigenvectrs related to the connection between reaactants and products are essential for the characterization of the mechanism. In our study, Reactants and products are linked through the potential energy curve

  1. Which kind of kinetic parameters are related with DE or DDE#? Are those values related with kinetic constants through Arrhenius equation? Please clarify such points.

  • R2: Thank you for your pertinent comment. In fact, the ΔE# and ΔΔE# values are directly related to the kinetic constants, serving as components of the Arrhenius equation. This equation is used to determine the reaction rate constant as a function of temperature. Specifically, ΔE# represents the activation energy, while ΔΔE# = ΔLIG2 - ΔLIG1 reflects the reactivity trend of QMPs, thereby avoiding the direct calculation of absolute energy values. These values are crucial for understanding reaction kinetics. The application of these terms was further clarified in the revised version of the manuscript. Now, a short explanation is described in the text.

  1. Regarding the % Resurrection linear adjustment, I think that authors need to use a larger data to conclude that a strong correlation is obtained, and to use such equation to calculate %Resurrection for new ligands.

  • R3: We acknowledge this important observation. As mentioned in the introduction of the manuscript, the scarcity of experimental studies in the literature restricts the available data. Only the studies by Zhuang and colleagues (2018) have successfully achieved the experimental resurrection of AChE, with a success rate of just 30% for the ligand C8. However, in line with the reviewer´s comment, we have introduced new compounds, Figure 1. Thus, new data, have been used to expand the dataset and evaluate the robustness of the correlation. The experimental results for these ligands were derived from the work of Zhuang and col-workers. The new results are described in Table 1 and 2.

ZHUANG, Q. G. et al. Demonstration of In Vitro Resurrection of Aged Acetylcholinesterase after Exposure to Organophosphorus Chemical Nerve Agents. Journal of Medicinal Chemistry, v. 61, n. 16, p. 7034 7042, Aug 2018. ISSN 0022-2623. Disponível em: <https://doi.org/10.1021/acs.jmedchem.7b01620>

Figure. 1 Structures of Realkylator Candidates.

Additionally, we introduced two ligands and performed the necessary calculations for these proposed compounds, as shown in Tables 2, 3 and 5. The results are thoroughly described and discussed in the text.

  1. Regarding the new ligands purposed, please justify the molecular modifications using a pharmacophoric model or an extensive discussion of the needed properties of the ligand based on the characteristic of the active site more than hydrophobicity, hydrogen bond interactions, pi-pi stacking, etc.

  • R4: Thank you for your constructive comment. In our study, the rationalization for molecular modifications of the new ligands is based on analyzing their interactions with the enzyme's active site and their % resurrection. We explore key molecular interactions, including hydrogen bonding, hydrophobic interactions, van der Waals forces, and π-π stacking. Additionally, ADME and toxicity studies were conducted theoretically, providing valuable insights into the pharmacokinetic and toxicological properties of the compounds. Regarding the modifications to ligands C4-1, C4-2, C4-3, and C4-4, these changes involve introducing double bonds and adding nitrogens to the five-membered ring, significantly impacting their properties and interactions with the aged acetylcholinesterase active site. The primary alterations concern the positions of double bonds and nitrogen atoms within the ring. Both C4-1 and C4-4 have a single nitrogen at position 1 of the ring; however, C4-1 includes two additional double bonds, whereas C4-4 has one. In contrast, C4-2 and C4-3 feature two double bonds and two nitrogen atoms. A crucial distinction between C4-2 and C4-3 lies in nitrogen positioning: in C4-2, they occupy positions 1 and 3, while in C4-3, they are at positions 1 and 2. These double bonds stabilize the ligand's conformation and enhance π-π interactions with aromatic residues in the active site, while added nitrogens promote hydrogen bonding. Moreover, nitrogens can alter the ring's electronic distribution, affecting binding affinity and interaction stability at the active site. For ligands C4-5 and C4-6, an OH group was introduced at different positions on the ring. This group acts as a hydrogen bond donor or acceptor, facilitating favorable interactions with charged residues in the active site and enhancing the ligand's affinity and specificity.

The presence of the OH group can restrict ligand flexibility, limiting its potential conformations within the active site. Favorable new conformations can increase affinity. These subtle structural modifications directly influence the electronic and geometric properties critical for optimizing ligand efficacy in the acetylcholinesterase resurrection process.

Our pharmacophore, ADME, and theoretical toxicity studies were pivotal in optimizing these ligands, ensuring that molecular adjustments led to compounds with favorable pharmacokinetic and toxicological profiles. We believe that combining active site interaction analysis with pharmacophore, ADME, and theoretical toxicity studies, although still evolving, provides a robust foundation for justifying the molecular modifications applied to these new ligands.

  1. In page 12 authors write:

“Furthermore, upon analyzing the differences in binding energy found in the 377 transition state of the ligand-enzyme complex, it can be observed that ligand C4 378 exhibited a smaller energy variation compared to the other ligands derived from its 379 structure”

What means a “transition state of the ligand-enzyme complex”? Seems that authors misunderstand the concept of transition state (https://doi.org/10.1021/jp953748q).

  • R5: We acknowledge this constructive comment. This commented sentence in page 12 considering the statement on page 12 concerning the "transition state of the enzyme-ligand complex," we agree that our description of the transition states was inadequate. We sincerely apologize for the incorrect usage of the term "transition state of the enzyme-ligand complex." It is clear that the correct definition pertains to the transition state of the chemical reaction itself, rather than the binding state of the ligand to the enzyme.

To clarify, our transition state analysis focuses on identifying and characterizing saddle points on the potential energy surface using computational methods based on density functional theory (DFT). Frequency analysis is employed to confirm the nature of the saddle point, involving a single imaginary mode corresponding to the reaction coordinate. The revised version of the manuscript will include a detailed explanation of how the transition states were computed, including associated chemical-structural changes, as well as the results of the frequency analysis.

Thank you again for your valuable comments and editorial assistance. We have worked to incorporate your suggestions and improve the quality of our manuscript.

Cordially,

Authors

Round 2

Reviewer 2 Report

Comments and Suggestions for Authors

Figure 1 needs to be improved to better quality

Please, explain better the low R² value found in Figures 5 and 6

In the phrase: “A strong correlation (R2 = 0.75) is observed between the theoretical and experimental values of % resurrection,  highlighting the importance of these steps for successful reactivation”. Please, show the data

Equation 1: Please change Ressureição to resurrection.

Please explain how the hydrophobicity degree of ligand C4 (Figure 7) influences the modifications.

References – Please provide the names of all authors in the reference list

Author Response

Revision notes

Dear Editors and Reviewers:

Thank you for your letter and the reviewers’ comments concerning our manuscript entitled “ Atomistic Origins of Resurrection of Aged Acetylcholinesterase by quinone methide precursors“. Those comments are all valuable and very helpful for revising and improving our paper, as well as the important guiding significance to our researches. We have studied comments carefully and have made corrections that we hope meet with approvals. Revised portions are marked in red in the manuscript. The main corrections in the paper and the responses to the reviewer’s comments are as following:

Responses to the Reviewers’ comments:

Figure 1 needs to be improved to better quality

  • R: Thank you for this valuable comment. In response to the reviewer's suggestion, we have added a new figure with better quality.

Please, explain better the low R² value found in Figures 5 and 6.

  • R: We acknowledge this important observation. In the revised manuscript, we have better explained the low R² values and their origins. The R² in Figure 5 represents the coefficient found when relating ΔE values from the docking study to the experimentally determined relative resurrection percentages. This low coefficient suggests that factors beyond interaction energy play a significant role in the resurrection. On the other hand, the R² in Figure 6 was found by relating the percentage of resurrection to ΔΔE# This low value indicates that kinetic parameters, specifically ΔΔE# alone, cannot adequately explain the experimental resurrection results. This suggests the need to incorporate other relevant effects associated with the binding process to achieve a better correlation between theory and experiment. Now, we have taken the opportunity to extend our discussion considering this point, pages 11 and 12.

In the phrase: “A strong correlation (R2 = 0.75) is observed between the theoretical and experimental values of % resurrection, highlighting the importance of these steps for successful reactivation”. Please, show the data.

  • R: Thank you very much for this. We acknowledge this important comment. According to the theoretical results presented in Equation 1, the association of the ligand with the aged enzyme and the subsequent chemical reaction are essential factors that significantly influence the AChE resurrection process. The equation developed to predict the percentage of resurrection for the investigated systems showed a good correlation (R² = 0.75) between the theoretical values (ΔE and ΔΔE#) and the experimental relative resurrection percentages, highlighting the importance of these steps for successful reactivation. The data used to find this correlation can be found in Table 2.

Equation 1: Please change Ressureição to resurrection.

  • R: Thank you for your observation, and we apologize for this mistake. The term 'Resurrection' has been added to Equation 1 in the revised manuscript.

Please explain how the hydrophobicity degree of ligand C4 (Figure 7) influences the modifications.

  • R: We appreciate this valuable comment. The revised manuscript text has been updated to include how the hydrophobicity of the C4 ligand influenced the modifications of the new proposed ligands. The degree of hydrophobicity can significantly affect both its interaction with the enzyme and its resurrection capacity. Hydrophobic ligands tend to bind more strongly to the hydrophobic regions of the enzyme, increasing the binding stability and the ligand's affinity for the active site. Additionally, hydrophobicity can impact the ligand's solubility and distribution in the biological environment, affecting its efficacy and specificity. The hydrophobicity of the ligand facilitates stronger van der Waals interactions with the enzyme's hydrophobic residues, which significantly contributes to the stability of the ligand-enzyme complex. Therefore, the search for an ideal ligand requires balancing hydrophobicity and polarity, considering its enzyme affinity, resurrection capacity, and potential toxicity during structural modifications.

References – Please provide the names of all authors in the reference list

  • We appreciate this comment. A thorough analysis of the references cited throughout the manuscript has been conducted.

Thank you very much for the excellent and professional revision of our manuscript. Furthermore, we thank the editorial assistance and hope with the changes and clarifications implemented, the manuscript would be now acceptable for publication in Molecules. Finally, we also remain at your disposal for any further inquiries.

Sincerely yours

Teodorico C. Ramalho

Department of Chemistry,

Federal University of Lavras,

Campus Universitário, C.P. 3037, 37200-000, Lavras,

Brazil.

          https://molecc.wixsite.com/molecc

          https://orcid.org/0000-0002-7324-1353
